# Targeted proteomics as a tool to detect SARS-CoV-2 proteins in clinical specimens

**Karel Bezstarosti**[1], **Mart M. Lamers**[2], **Wouter A. S. Doff**[1], **Peter C. Wever**[3], **Khoa T. D. Thai**[ID][4], **Jeroen J. A. van Kampen**[2], **Bart L. Haagmans**[2], **Jeroen A. A. Demmers**[ID][1]*

**1** Proteomics Center, Erasmus University Medical Center, Rotterdam, the Netherlands, **2** Viroscience Department, Erasmus University Medical Center, Rotterdam, the Netherlands, **3** Jeroen Bosch Hospital, 's-Hertogenbosch, the Netherlands, **4** Star-shl Medical Diagnostic Center, Rotterdam, the Netherlands

* j.demmers@erasmusmc.nl

**Data Availability Statement:** Mass spectrometry data files have been deposited in the ProteomeXchange repository with identifier

## Abstract

The rapid, sensitive and specific detection of SARS-CoV-2 is critical in responding to the current COVID-19 outbreak. In this proof-of-concept study, we explored the potential of targeted mass spectrometry (MS) based proteomics for the detection of SARS-CoV-2 proteins in both research samples and clinical specimens. First, we assessed the limit of detection for several SARS-CoV-2 proteins by parallel reaction monitoring (PRM) MS in infected Vero E6 cells. For tryptic peptides of Nucleocapsid protein, the limit of detection was estimated to be in the mid-attomole range (9E-13 g). Next, this PRM methodology was applied to the detection of viral proteins in various COVID-19 patient clinical specimens, such as sputum and nasopharyngeal swabs. SARS-CoV-2 proteins were detected in these samples with high sensitivity in all specimens with PCR Ct values <24 and in several samples with higher CT values. A clear relationship was observed between summed MS peak intensities for SARS-CoV-2 proteins and Ct values reflecting the abundance of viral RNA. Taken together, these results suggest that targeted MS based proteomics may have the potential to be used as an additional tool in COVID-19 diagnostics.

## Introduction

Severe acute respiratory syndrome coronavirus 2 (SARS-CoV-2) is the causative agent of coronavirus disease 2019 (COVID-19), which is a severe respiratory disease [1]. The World Health Organization (WHO) has designated the ongoing pandemic of COVID-19 a Public Health Emergency of International Concern [2]. As of now, over one million deaths have been reported worldwide and this is probably an underestimation because of lack of testing capacity in large parts of the world.

SARS-CoV-2 is a positive-sense single-stranded RNA virus, which encodes several non-structural proteins such as spike, envelope, membrane and nucleocapsid protein [3]. Rapid, sensitive and specific diagnosis of SARS-CoV-2 is widely recognized to be critical in responding to this outbreak, but also for long-term improvements in patient care. Importantly, the reduction of time required to identify SARS-CoV-2 infections will significantly contribute to

PXD025294 (https://www.ebi.ac.uk/pride/archive/
projects/PXD025294).

**Funding:** The author(s) received no specific
funding for this work.

**Competing interests:** The authors have declared
that no competing interests exist.

**Abbreviations:** PRM, parallel reaction monitoring;
AUC, area under the curven; LC-MS, nanoflow
liquid chromatography–mass spectrometry; Ct,
threshold value; PCR, polymerase chain reaction.

limiting the enormous social and economic consequences of this large global society paralyzing outbreak. Conventional methods for diagnostic testing of viral infections, which are also widely used for SARS-CoV-2 testing, are based on polymerase chain reaction (PCR) or other (multiplexed) nucleic-acid based technologies and antigen detection. Since its emergence late 2019 it has become clear that additional diagnostic tools that target SARS-CoV-2 should be developed to complement existing tools in a "proactive approach" proposed by the *Coronaviridae Study Group of the International Committee on Taxonomy of Viruses* [1]. Alternative and/ or complementary SARS-CoV-2-specific diagnostic tests are desperately needed since the current testing capacity is insufficient, amongst others because of shortages of supplies such as RNA extraction kits, PCR reagents and delivery issues for primers and probes.

Besides PCR based approaches, immunoassays have been employed in the detection of other viruses. In addition, mass spectrometry (MS) based techniques have been applied previously, for instance to detect influenza virus proteins [4] and human metapneumovirus (HMPV) in clinical samples [5]. Recent developments in targeted proteomics methods and Orbitrap mass spectrometry such as parallel reaction monitoring (PRM) have shown a substantial sensitivity increase. Although mass spectrometry based approaches have been used in several SARS-CoV-2 studies [6–10] (and reviewed in [11 and 12]), is not yet clear whether state-of-the-art proteomics technologies could provide the sensitivity and specificity needed in diagnostics.

Here, we explore the use of targeted mass spectrometry based proteomics for SARS-CoV-2 detection in research and clinical samples. For this, we first assessed the limit of detection by parallel reaction monitoring (PRM) on an Orbitrap mass spectrometer for specific tryptic peptides of SARS-CoV-2 proteins. The sensitivity was found to be in the mid-attomole range (~9.0E13 g) for Nucleocapsid protein. Next, we sought whether this sensitivity is sufficient for the detection of SARS-CoV-2 in clinical specimens such as nasopharyngeal swabs, mucus and sputum. This largely depends on the absolute amounts of viral proteins as well as on the complexity and abundance of the proteinaceous matrix background present in such samples. Using PRM, we could indeed detect various proteolytic peptides of several SARS-CoV-2 proteins in sputum and swab samples. In different cohorts of individuals tested positive for COVID-19, using this PRM MS we were able to detect and relatively quantify SARS-CoV-2 tryptic peptides. Moreover, we observed a clear relationship between the peak intensities in the mass spectra and the Ct (threshold cycle) values obtained from PCR assays of the same samples. For all samples with Ct values of up to ~24, tryptic peptides were detected and quantified. Even for several samples with higher Ct values, SARS-CoV-2 peptides could reliably be detected. In addition, we have explored several methods to increase the sensitivity of the method even further and to decrease the sample analysis times.

In conclusion, this proof-of-concept study shows that the sensitivity of targeted proteomics is sufficiently high for the detection of viral material in patient samples such as swabs, sputum, mucus and suggests that other types of body fluids can be used as source material. The method that we describe here can be transferred to clinical diagnostic labs that host mass spectrometry equipment. Subsequent steps should be focused on sample preparation protocols that are in agreement with validated virus inactivation procedures, improvements in sample throughput and increase in sensitivity of detection.

Finally, providing novel mass spectrometry based diagnostic tools that complement genomic approaches is also the major goal of the recently formed COVID-19 mass spectrometry coalition (www.covid19-msc.org). The aim of this proof-of-concept study is to highlight the potential of mass spectrometry in identifying SARS-CoV-2 proteins for diagnostics and research.

## Materials and methods

### Virus and cells

Vero E6 cells were maintained in Dulbecco's modified Eagle's medium (DMEM, Gibco) supplemented with 10% fetal calf serum (FCS), HEPES, sodium bicabonate, penicillin (final concentration 100 IU/mL) and streptomycin (final concentration 100 IU/mL) at 37˚C in a humidified $CO_2$ incubator. SARS-CoV-2 (isolate BetaCoV/Munich/BavPat1/2020; European Virus Archive Global #026V-03883; kindly provided by Dr. C. Drosten) was propagated on Vero E6 cells in Opti-MEM I (1X) + GlutaMAX (Gibco), supplemented with penicillin (final concentration 100 IU/mL) and streptomycin (final concentration 100 IU/mL) at 37˚C in a humidified $CO_2$ incubator. Stocks were produced by infecting cells at a multiplicity of infection (MOI) of 0.01 and incubating the cells for 72 hours. The culture supernatant was cleared by centrifugation and stored in aliquots at −80˚C. Stock titers were determined by preparing 10-fold serial dilutions in Opti-MEM I (1X) + GlutaMAX. Aliquots of each dilution were added to monolayers of 2E04 VeroE6 cells in the same medium in a 96-well plate. Plates were incubated at 37˚C for 5 days and then examined for cytopathic effect. The TCID50 was calculated according to the method of Spearman & Kärber. All work with infectious SARS-CoV and SARS-CoV-2 was performed in a Class II Biosafety Cabinet under BSL-3 conditions at Erasmus University Medical Center.

### Organoid-derived human airway culture secretions

Organoid-derived human airway culture secretions were harvested from cultures that had been differentiated at air-liquid interphase for 3 weeks as described by Lamers *et al.* [13]. Secretions could be harvested by pipetting using a P1000 tip and were not diluted. Secretions were stored at -80˚C until use. Ten-fold dilutions of virus stock containing 1.21E06 TCID50/ml were made in Opti-MEM I (1X) + GlutaMAX. Next, 25 µl of each virus dilution was mixed with 25 µl of airway culture secretions. Virus was inactivated by adding 50 µl of 2X Laemmli buffer (BioRad) and incubating at 95˚C for 10 minutes.

### Collection and treatment of patient material samples

Nasopharyngeal swabs from COVID-19 patients were stored in universal transport medium (UTM; contains bovine serum albumin) after collection. Next, they were centrifuged at 15,000 g for 3 min to pellet down cell debris (termed 'swab pellet'). The swabs were then washed twice with PBS to remove excessive albumin and fixed in 80% acetone (termed 'swab supernatant'). Sputum from COVID-19 patients was collected and diluted in UTM. Alternatively, sputum was diluted in medium after collection and a few droplets were pipetted on glass slides, dried and fixed in 80% acetone.

   The nasopharyngeal and throat swabs and sputum samples were obtained from different patients. Samples of sputum deposited on glass slides were obtained from one single patient.

   Proteins present in patient nasopharyngeal and throat swabs or sputum samples in transport medium were first precipitated with acetone-TCA to remove excessive albumin according to [14]. Briefly, 40 µl of the sample was mixed with 400 µl acetone and 1% TCA and left overnight at -20˚C. Proteins were pelleted, washed once with ice-cold acetone and left to dry for 5 min. The protein pellet was then resuspended in 40 µl 50 mM Tris/HCl, 4 M urea (pH 8.2) and diluted with 160 µl 50 mM Tris/HCl (pH 8.2).

   Cellular human and viral material in sputum deposited on glass slides was lysed in 50 µl 2% SDS dissolved in 50 mM Tris/HCl (pH 8.2) followed by sonication in a Bioruptor Pico (Diagenode). Proteins were digested using the SP3 protocol as described below.

## Sample preparation for MS

A 90% confluent T75 flask of VeroE6 was infected at a MOI of 0.3 and incubated for 24 hours at 37˚C in a humidified $CO_2$ incubator. Next, cells were collected by scraping and the medium was removed after centrifuging at 400 g for 5 min. Cells were lysed in 2X Laemmli buffer (final concentration; Bio-Rad) and boiled at 95˚C for 20 min to inactivate the virus. Proteins were reduced and alkylated with DTT (Sigma) and IAA (Sigma) and precipitated using chloroform/methanol [15]. The protein pellet was then dissolved in 100 μl of a 50 mM Tris/HCl buffer (pH 8.0) with 2 M urea. Proteins were quantified using the BCA protein kit (ThermoFisher Scientific / Pierce, #23225); peptides were quantified with a quantitative colorimetric peptide assay (ThermoFisher Scientific / Pierce, #23275). Fifty μg of protein was digested with 1 μg trypsin (Thermo) overnight at room temperature. The peptide digest was cleaned on a 50 mg tC18 Sep-Pak cartridge (Waters) and the peptides were eluted with 2 ml acetonitrile/water (1:1) with 0.05% TFA.

Alternatively, proteins were digested with trypsin using the SP3 protocol [16], with minor modifications. Briefly, proteins in 30 μl Laemmli buffer were reduced for 30 min at 50˚C with 5 mM DTT and alkylated with 10 mM IAA. A slurry of 10 μg of Sera-Mag speedbeads (GE Healtcare) in 20 μl milliQ/ethanol (1:1, vol/vol) was added to the solution and mixed for 10 min at RT. Using a magnetic rack, the beads were immobilized and washed three times with 100 μl 80% ethanol. 1 μg trypsin and 0.5 μg Lys-C in 100 μl 50 mM Tris/HCl pH 8.3 were added to the beads and the sample was incubated overnight at 37˚C. The tryptic digest was then acidified with TFA and desalted using a StageTip. Peptides were eluted with 100 μl 40% acetonitrile and 0.1% formic acid and dried using a Speedvac. Before analysis by LC-MS peptides were dissolved in 20 μl 2% acetonitrile / 0.1% formic acid.

For PRM measurements, peptide samples with concentrations ranging from 0 to 25 ng/μl were prepared from SARS-CoV-2 infected VeroE6 cell lysates. For global proteomics, peptides were fractionated off-line using high pH reversed-phase (ThermoFisher / Pierce, #84868) into four fractions.

Synthetic AQUA peptide analogs containing a heavy stable isotope labeled C-terminal Arginine (R10) residue were purchased from Thermo.

## LC-MS

Peptide mixtures were trapped on a 2 cm x 100 μm Pepmap C18 column (ThermoFisher Scientific, #164564) and separated on an in-house packed 50 cm x 75 μm capillary column with 1.9 μm Reprosil-Pur C18 beads (Dr. Maisch) at a flow rate of 250 nL/min on an EASY-nLC 1200 (ThermoFisher Scientific), using a linear gradient of 0–32% acetonitrile (in 0.1% formic acid) during 60 or 90 min. The eluate was directly sprayed into the mass spectrometer by means of electrospray ionization (ESI).

For targeted proteomics, a parallel reaction monitoring regime (PRM) was used to select for a set of previously selected peptides on an Orbitrap Eclipse Tribrid mass spectrometer (ThermoFisher Scientific) operating in positive mode and running Tune version 3.3. Precursors were selected in the quadrupole with an isolation width of 0.7 m/z and fragmented with HCD using 30% collision energy (CE). MS2 spectra were recorded in profile mode in the Orbitrap at 30,000 resolution. The maximum injection time was set to dynamic with a minimum of 9 points measured across the chromatographic peak. See **S2 Fig** for the isolation list. For global DDA proteomics, data were recorded on an Orbitrap Fusion Lumos Tribrid mass spectrometer (ThermoFisher Scientific) in data dependent acquisition (DDA) mode. Full MS1 scans were recorded in the range of 375–1,400 *m/z* at 120,000 resolution. Fragmentation of peptides with charges 2–5 was performed using HCD. The collision energy was set at 30% and

previously fragmented peptides were excluded for 60 seconds. The resolution of tandem mass spectra (MS2) was set at 30,000 and automatic gain control (AGC) was set to 5E4 and the maximum injection time (IT) was set to 50 ms. MS2 spectra were recorded in centroid mode. The sequence of sampling was blanks first and then in order of increasing peptide input amounts to avoid any contamination of previous samples.

### Data analysis

Mass spectrometry data were analyzed using Mascot v 2.6.2 within the Proteome Discoverer v 2.3 (PD, ThermoFisher Scientific) framework or with MaxQuant v 1.6.10.43 (www.maxquant. org), all with standard settings (note: fragment tolerance set to 20 ppm). Raw data recorded on the Orbitrap Eclipse with the FAIMS option were first converted into mzXML format using the FAIMS MzXML Generator software tool (Coon Lab) before MaxQuant analysis. PRM data were analyzed with Skyline (skyline.ms). Spectra and chromatograms were visualized in PD 2.3, Skyline or the PDV proteomics viewer (pdv.zhang-lab.org). The Skyline output was converted to ridgeline plots using in-house developed software. Calibration curves were generated for several endogenous tryptic peptides of SARS-CoV-2 proteins from infected Vero E6 cell samples. The Skyline settings were: regression fit: bilinear through zero; regression weighting: none; normalization method: none; LOD calculation method: bilinear turning point: mac LOQ bias: [left empty]; max LOQ CV: [left empty]; qualitative ion ratio threshold: [left empty]). For global proteome analyses the UniprotKB SARS2 database (https://covid-19.uniprot.org/; 14 entries; May 2020) was concatenated with the UniprotKB database, taxonomy *Chlorocebus* (African green monkey) or taxonomy *Homo sapiens* (version Oct 2019).

### Ethics statement

The Institutional Review Board of both the Jeroen Bosch Hospital and the Star-shl Medical Diagnostic Center approved this study. The boards approve anonymous use of remnant biospecimens for scientific purposes. All patients were informed of the possibility that residual samples could be used anonymously for research purposes with right of refusal. All samples were anonymized before they arrived at the Erasmus MC Proteomics Center for further analysis.

## Results and discussion

We set off by analyzing the global proteome of Vero E6 cells infected with SARS-CoV-2 using standard bottom-up proteomics. Upon off-line high pH reversed-phase (RP) peptide fractionation, LC-MS was performed on an Orbitrap Lumos and RAW files were combined during data analysis. SARS-CoV-2 proteins were measured with high sequence coverage as exemplified in **Fig 1** and S1 **and S2 Figs**. Based on a label free semi-quantitative (LFQ) analysis of MaxQuant output data, we estimate that 4–5% of the total proteome of this sample (composed of Vero cells, viral proteins inside cells and viral particles outside of cells in the supernatant) is made up of viral proteins. Of all SARS-CoV-2 proteins covered Nucleocapsid is the most abundant one, making up > 88% of all signal intensity as calculated from MaxQuant intensity values (**Table 1**). Therefore, if intensity values can be used as a proxy for total protein abundance, almost 90% of the SARS-CoV-2 proteome would consist of Nucleocapsid. Abundance of the Nucleocapsid protein in the samples is due to the high level production of this protein in cells as a result of the nested set of mRNAs produced during replication and the resulting overproduction of this protein. Moreover, the high number of identified *Chlorocebus* proteins (>6,000; see **S1 File**) suggests that it is possible to not only study SARS-CoV-2 proteins, but to also investigate the effects of viral infection on the host cell proteome in great detail.

| Source | # identified proteins | # identified peptides |
|--------|----------------------|----------------------|
| Total | 6,512 | 61,394 |
| SARS2 | 9 | 279 |
| *Chlorocebus* | 6,503 | 61,115 |

| Accession | Protein name | Organism | Description | Coverage [%] | # Peptides | # PSMs |
|-----------|-------------|----------|-------------|-------------|-----------|--------|
| P0DTC9 | NCAP | SARS2 | Nucleoprotein OS=Severe acute respiratory syndrome coronavirus 2 OX=2697049 GN=N PE=3 SV=1 | 91 | 61 | 676 |
| P0DTD1 | R1AB | SARS2 | Replicase polyprotein 1ab OS=Severe acute respiratory syndrome coronavirus 2 OX=2697049 GN=rep PE=1 SV=1 | 22 | 128 | 237 |
| P0DTC2 | SPIKE | SARS2 | Spike glycoprotein OS=Severe acute respiratory syndrome coronavirus 2 OX=2697049 GN=S PE=1 SV=1 | 39 | 53 | 172 |
| P0DTD2 | ORF9B | SARS2 | Protein 9b OS=Severe acute respiratory syndrome coronavirus 2 OX=2697049 PE=3 SV=1 | 96 | 12 | 49 |
| P0DTC5 | VME1 | SARS2 | Membrane protein OS=Severe acute respiratory syndrome coronavirus 2 OX=2697049 PE=3 SV=1 | 36 | 13 | 89 |
| P0DTC3 | AP3A | SARS2 | Protein 3a OS=Severe acute respiratory syndrome coronavirus 2 OX=2697049 GN=3a PE=3 SV=1 | 28 | 8 | 30 |
| P0DTC7 | NS7A | SARS2 | Protein 7a OS=Severe acute respiratory syndrome coronavirus 2 OX=2697049 GN=7a PE=3 SV=1 | 23 | 1 | 2 |
| P0DTC6 | NS6 | SARS2 | Non-structural protein 6 OS=Severe acute respiratory syndrome coronavirus 2 OX=2697049 GN=6 PE=3 SV=1 | 31 | 2 | 2 |
| P0DTC4 | VEMP | SARS2 | Envelope small membrane protein OS=Severe acute respiratory syndrome coronavirus 2 OX=2697049 GN=E PE=3 SV=1 | 16 | 1 | 2 |

**Fig 1. Numbers of identified proteins in SARS-CoV-2 infected Vero cells (PD2.3/Mascot search engine, offline high pH RP fractionation into four fractions, total input material 0.6 μg, 90 min LC gradients on an Orbitrap Lumos).**

MaxQuant output of LFQ analysis of SARS-CoV-2 infected Vero E6 cell lysates. Intensity values were taken directly from the MaxQuant ProteinGroups.txt output file. The indicated percentage is relative to the global viral proteome.

Based on the extensive sequence coverage for Nucleocapsid and several other SARS-CoV-2 proteins we established a list of peptide targets that can be used for PRM targeting. These molecular finger prints are used to program the mass spectrometer in such a way that it acts as a filter to let only those specific SARS-CoV-2 proteolytic fragments pass. This way, a specific set of target peptides/proteins can be searched for in basically any sample from which proteins can be isolated (*e.g.*, in vitro cell cultures, patient derived samples, etc.).

Three highly mass spectrometric responsive tryptic peptides were selected from the global proteome data set as targets for PRM, i.e. GFYAEGSR (NCAP_SARS2), ADETQALPQR (NCAP_SARS2) and EITVATSR (VME1_SARS2). Importantly, there are potentially a few dozens of specific SARS-CoV-2 peptides that could be used for targeting, although some of these may show slightly lower mass spectrometric responsiveness.

**Table 1. SARS-CoV-2 protein composition.**

| Protein | Intensity | Percentage | MW (kDa) |
|---------|-----------|-----------|----------|
| P0DTC9|NCAP_SARS2 | 3.20E+11 | 88.6 | 46 |
| P0DTC2|SPIKE_SARS2 | 1.77E+10 | 4.9 | 141 |
| P0DTC5|VME1_SARS2 | 1.36E+10 | 3.8 | 25 |
| P0DTD1|R1AB_SARS2 | 4.61E+09 | 1.3 | 794 |
| P0DTD2|ORF9B_SARS2 | 2.91E+09 | 0.8 | 11 |
| P0DTC3|AP3A_SARS2 | 1.94E+09 | 0.5 | 31 |
| P0DTC6|NS6_SARS2 | 3.08E+08 | 0.1 | 7 |
| P0DTC4|VEMP_SARS2 | 4.96E+07 | 0.0 | 8 |

Our test sample, *i.e.* Vero E6 cells infected with SARS-CoV-2, contained 2.0 mg/ml protein based on a BCA assay. The results of the colorimetric peptide quantification after digestion were in agreement with this concentration. A dilution series was prepared from this sample and the injected total peptide quantities ranged from 50 ng down to 20 pg. These extensively diluted samples were then subjected to PRM on an Orbitrap Eclipse and the areas under the curve (AUCs) were used for target peptide quantitation. **Fig 2** shows the results of this PRM assay. The six most intense (Top6) fragment ion peaks are shown in different colors as overlapping (in terms of retention time) peaks. The chromatogram excerpts are shown from top to bottom and left to right for decreasing total protein input concentrations. The lower right chromatogram in each panel shows the Top6 fragment ions in the sample corresponding to 20 pg total protein input, which could thus be regarded as the limit of detection (LOD). It should be noted that all PRM assays are performed on peptide targets that are present in a complex matrix, *i.e.* a Vero cell lysate.

## Detection of SARS-CoV-2 peptides in clinical specimens

As a proof-of-concept experiment we then applied this targeted proteomics technology to detect SARS-CoV-2 proteins in samples from COVID-19 patients. Several different types of patient samples were collected and provided to us by the Erasmus MC diagnostic department. Since all viral infectivity in these clinical specimens needs to be abolished according to established protocols in an BSL-3 facility before they can be further processed, the condition of the starting material was not optimized for subsequent proteomics. Notably, some clinical samples contained high amounts of contaminants such as detergents, albumin, etc. Sputum diluted in viral transport medium deposited on glass slides and then simply fixed in 80% acetone turned out to be the sample type that was most compatible with the subsequent proteomics workflow. Apparently, the relatively simple background matrix composition combined with a sample preparation protocol that does not involve the addition of detergents or albumin offers a substantial advantage for proteomics workflows.

For PRM, we focused on the SARS-CoV-2 Nucleocapsid tryptic peptide AYNVTQAFGR, since this peptide was found to be one of the most prominent and responsive peptides in the SARS2-CoV-2 infected Vero E6 cell lysate. Also, this amino acid sequence is unique to SARS-CoV-2, even in comparison to SARS-CoV.

In order to unambiguously confirm the presence of Nucleocapsid peptides we compared the chromatogram patterns of AYNVTQAFGR with those of a variant of this peptide that contains a heavy isotope labeled C-terminal Arginine. This synthetic AQUA peptide was spiked in all patient samples and co-elutes with the corresponding (non-labeled) endogenous peptide in LC-MS because of its similar biophysical properties. For a selection of clinical samples the reconstructed Skyline PRM chromatograms of the target peptides are shown in **Fig 3A** (sputum) and **3B** (swab). For all samples, the endogenous peptides are shown in the lower panels, while the corresponding AQUA counterpart peptides are shown in the upper panels. The similarities in both fragment ion chromatogram pattern and elution time confirm the presence of SARS-CoV-2 proteins in all sputum samples and sample 'swab supernatant 4'.

To investigate the relationship between amounts of viral RNA as detected by PCR methods and protein abundances determined by mass spectrometric methods, we collected two cohorts of clinical specimens with known PCR Ct values ranging from the 12 to >30. These samples were nasopharyngeal Eswabs, Aptima or Sigma swabs from individuals who had tested positive for COVID-19 in regular diagnostic assays. The viral material in these swabs was first inactivated in 80% acetone Swabs and similar proteomics sample preparation procedures were followed as for the sputum samples described earlier. For several target tryptic peptides of

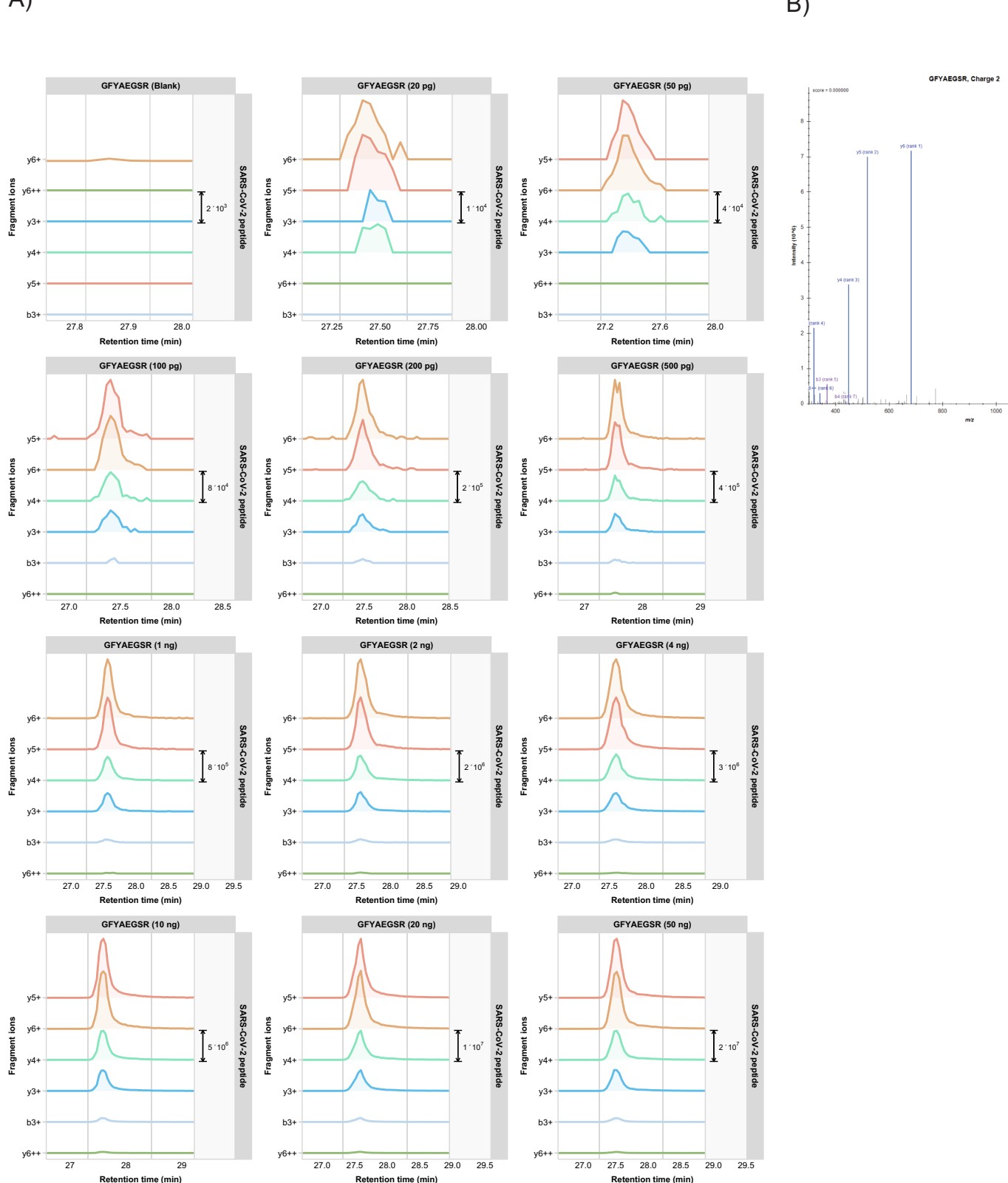

**Fig 2. PRM results visualized in skyline (skyline.ms).** Chromatograms for each of the Top6 fragment ions are shown in different colors in a dilution series for tryptic peptides A) GFYAEGSR (NCAP_SARS2), B) ADETQALPQR (NCAP_SARS2) and C) EITVATSR (VME1_SARS2). The lower right chromatogram represents the lowest sample input, *i.e.* 20 pg. The MS/MS spectrum on the right is the library spectrum. C) Calibration curves based on PRM data for three target peptides recorded on an Orbitrap Eclipse. The summed AUC values for the Top6 fragment ions of each peptide were taken for relative quantitation. 'Input' is *total* protein input from the SARS2-CoV-2 infected Vero E6 cell lysate; inserts are zoom-ins of the input range 0–300 pg.

SARS-CoV-2 proteins, AQUA peptide counterparts were included in the samples as spike-in. Relative protein abundances were defined by the sum of the AUCs for all fragment ion chromatograms for every peptide of each viral protein detected in a sample.

For the first patient cohort, the mass spectrometry data are shown in **Fig 4A** and **S7 File**. For all specimens with E-Gen CT value <20 relatively high mass spectral peak intensities were observed for various target peptides in our PRM assay. Also, for several specimens with Ct values in the low 20s viral protein could still be unambiguously detected. For example, in sample #5 peptide GQGVPINTNSSPDDQIGYYR was identified by eight highly mass accurate fragment ions (**Fig 4B**). The correlation between the PCR Ct values and the summed mass spectrometry intensities is shown in **Fig 4C**. There is a clear inverse relationship between these

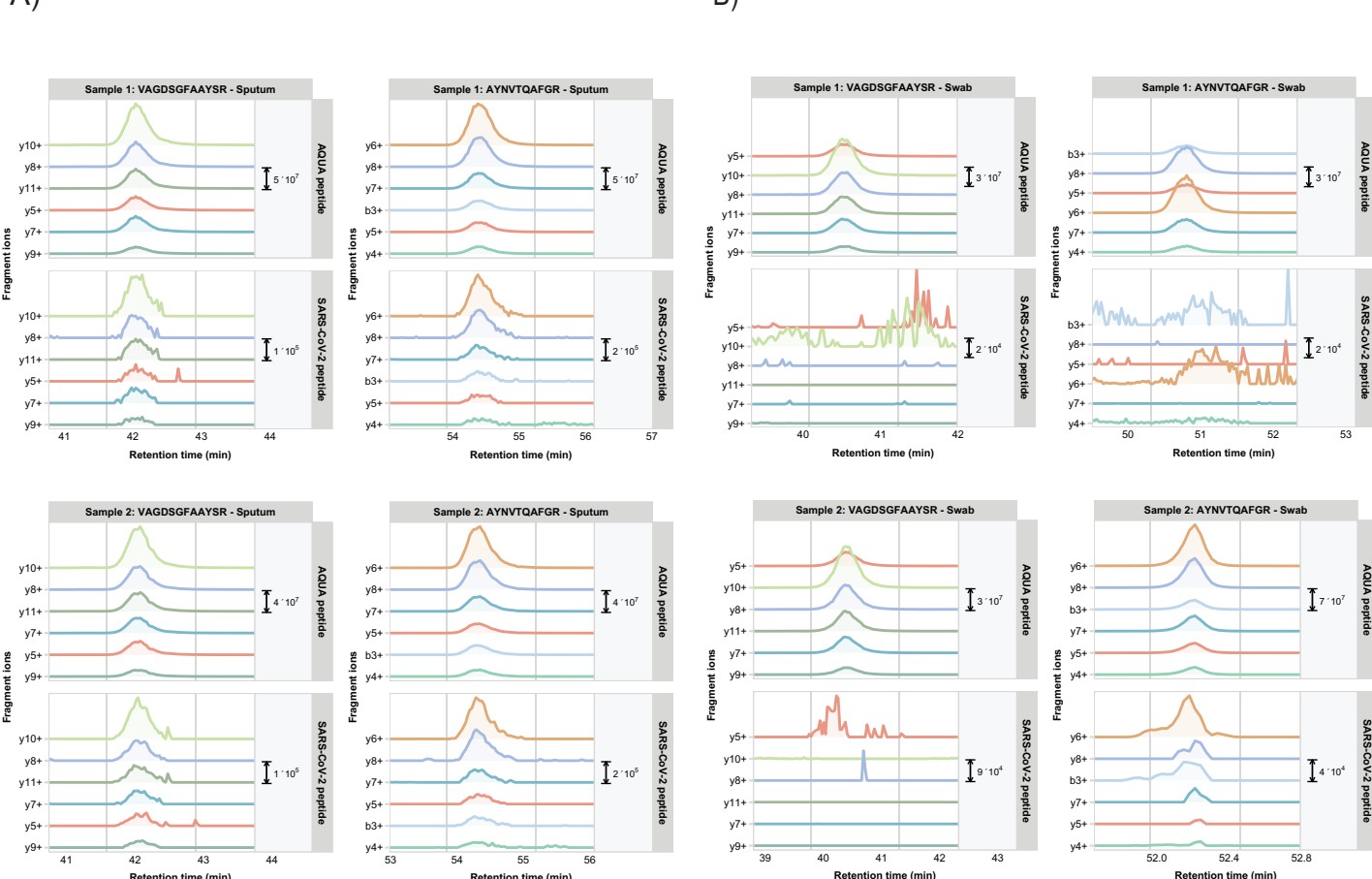

**Fig 3. PRM fragment ion chromatograms of SARS-CoV-2 Nucleocapsid and VME1 tryptic peptides VAGDSFAAYSR and AYNVTQAFGR in representative A) sputum specimens and B) throat swab specimens of COVID-19 patients.** Chromatograms for each of the Top6 fragment ions are shown in different colors. The upper panels show the fragment ion chromatograms of the corresponding synthetic AQUA peptides VAGDSFAAYS[R] (*m/z* 605.79) and AYNVTQAFG[R] (*m/z* 568.79). See S4 Fig for additional clinical specimens.

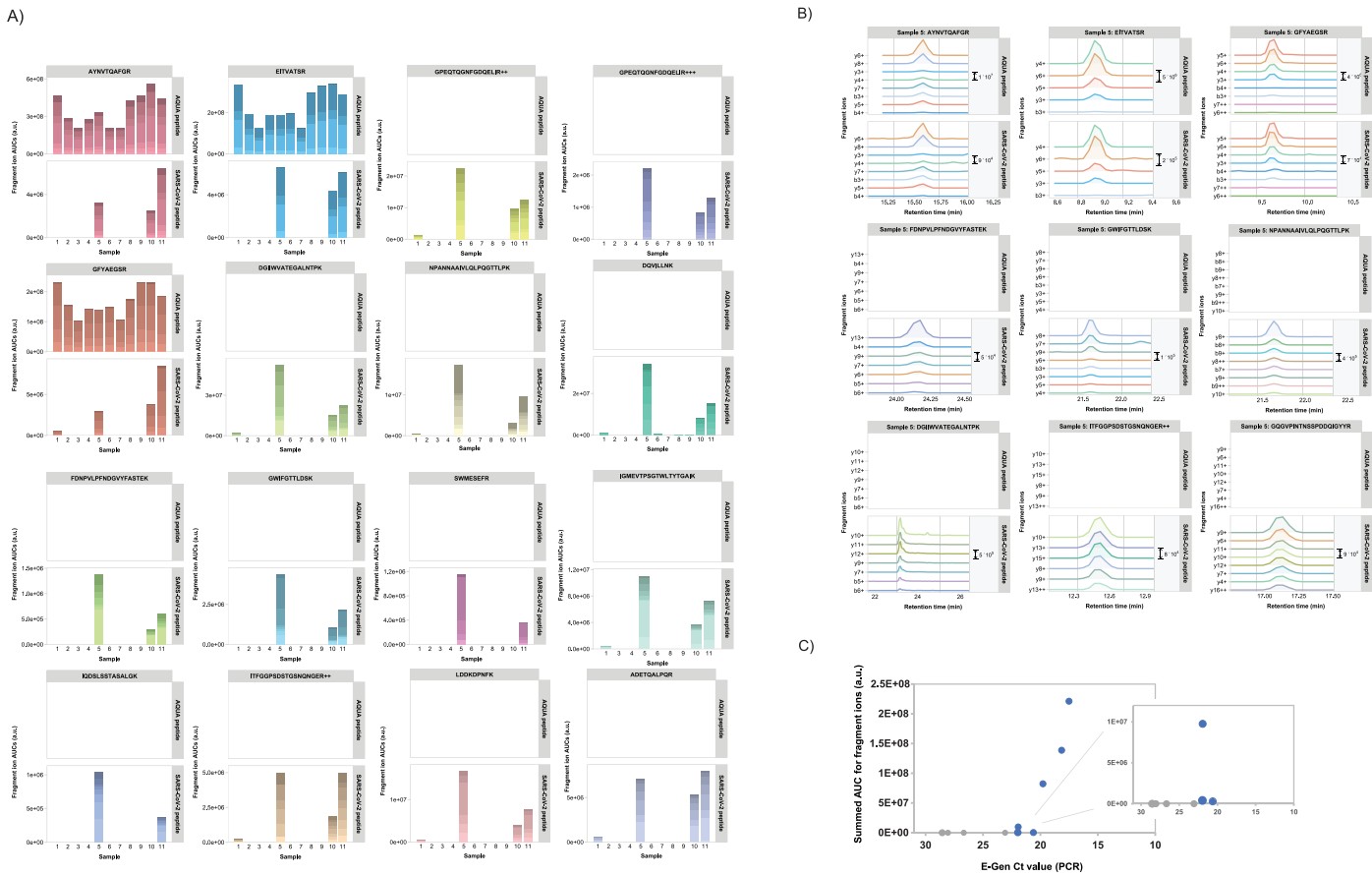

**Fig 4. PRM data of clinical specimens of COVID-19 patients (cohort 1).** A) Total AUCs of SARS-CoV-2 target peptide fragment ion chromatograms (upper panels show the spiked-in AQUA peptide signals; if no AQUA peptide counterpart was available upper panels are left empty). The color shading of the bars indicate the relative AUCs of the different fragment ions. B) Fragment ion chromatograms for various SARS-CoV-2 target peptides in one representative clinical specimen. C) Comparison of AUCs versus PCR Ct values for clinical specimens. Data points in grey represent samples in which no target peptides were detected by PRM.

sample characteristics, with a threshold value for detection by targeted mass spectrometry around Ct value 22. For some samples with high Ct values no SARS-CoV-2 peptides could be detected.

The second sample cohort consisted of 15 nasopharyngeal swabs from individuals who had tested positive for COVID-19. Since the two sample cohorts were collected and analyzed at different COVID-19 diagnostic testing sites, the results cannot be directly compared to one another. For this reason, we treat them separately here.

Viral samples were collected with different types of collection kits, such as Eswabs, Aptima and Sigma swabs, and the same sample preparation procedures were followed as for the sputum samples described earlier.

Positive mass spectrometric detection was observed for all specimens with E-Gen CT values <24 (**Fig 5**), although the absolute summed intensities of target peptide fragments varied widely. Individual peptide identifications and quantifications are visualized in (**Fig 5A and 5B** and **S8 File**) and compared to corresponding AQUA peptide counterparts if applicable. Clearly, the highest summed AUCs values were observed for samples 1, 8 and 14, which have very low PCR CT values and thus a relatively high amount of viral material. Strikingly, for several patient samples different sets of target peptides were more pronouncedly detected,

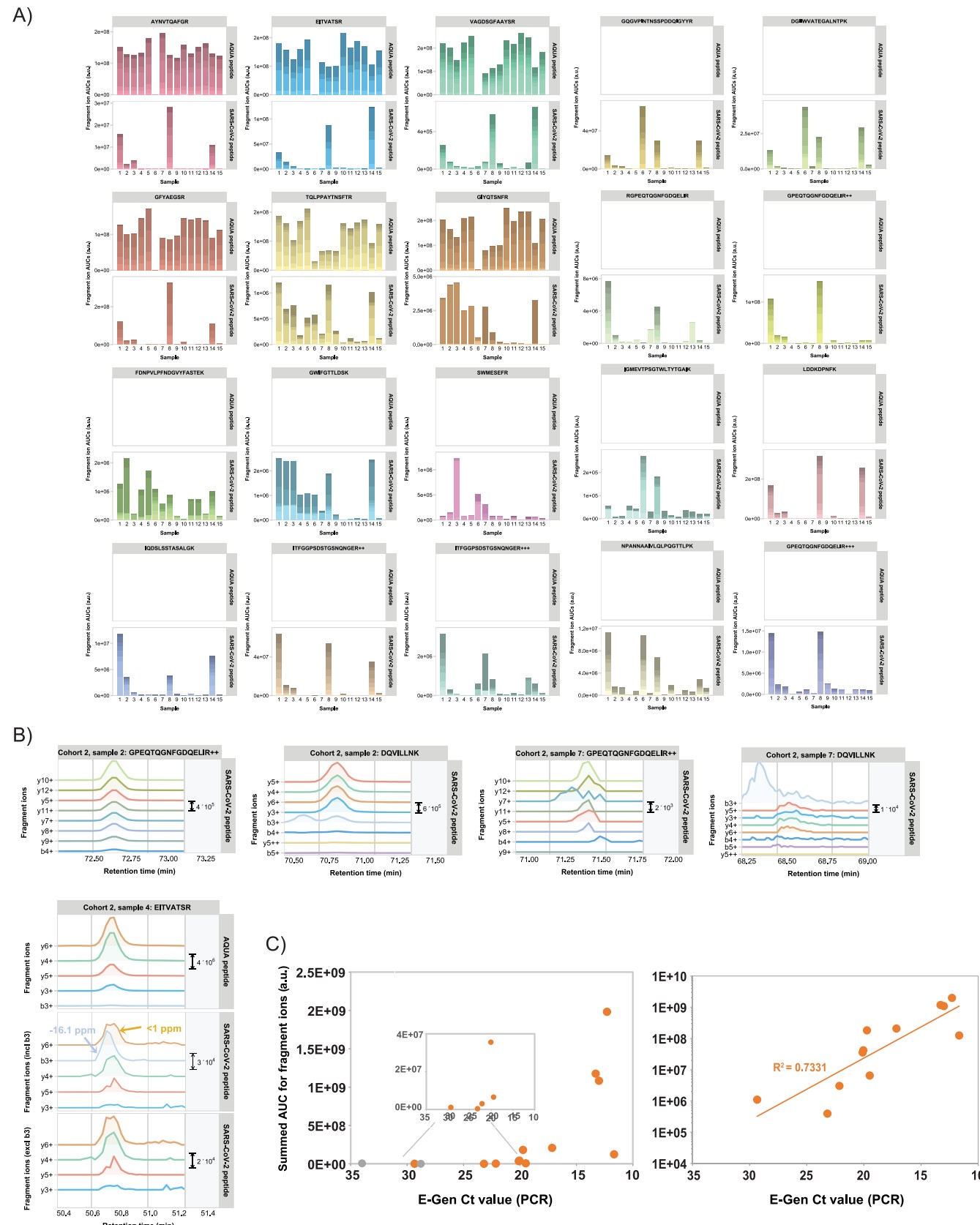

**Fig 5. PRM data of clinical specimens of COVID-19 patients (cohort 2).** A) Total AUCs of SARS-CoV-2 target peptide fragment ion chromatograms (upper panels show the spiked-in AQUA peptide signals; if no AQUA peptide counterpart was available upper panels are left empty). The color shading of the bars indicate the relative AUCs of the different fragment ions. B) Selection of PRM results for several target peptides in samples 2, 4 and 7. See main text for explanation. Retention times for the identical peptides in different samples may slightly differ as a result of small variations in LC gradients and chromatography setup. C) Comparison of AUCs versus PCR Ct values for clinical specimens. Data points in grey represent samples in which no target peptides were detected by PRM.

although the exact same sample preparation protocols were followed for all samples. This may reflect the heterogeneity of the samples, possibly leading to quite diverse outcomes of protein digestion procedures. For example, in sample #6 several relatively long peptides are highly abundant, while some shorter ones are virtually absent. Whether or not the differences in detected target peptide sets may reflect the status of the viral particle (*e.g.* active particles versus aggregated, non-assembled proteins from viral residue) is the subject of current research.

Overall, there is an inverse relationship ($R^2$ = 0.733) between $^{10}$Log transformed summed AUCs of the fragment ion chromatograms and the E-Gen Ct values from PCR assays on the same samples (**Fig 5D**), which makes sense because of the logarithmic nature of the Ct value scale and which reflects the amounts of virus RNA and proteins present. Obviously, the number of data points is only limited in our case and the strength of this relationship is expected to become stronger with an increasing number of data points.

Two target peptides were detected in a sample with a Ct value of 23.2 (**Fig 5C**; sample #7, peptides GPEQTQGNFGDQELIR and DQVILLNK. Sample #2 is shown for comparative purposes). Strikingly, for one specimen with a Ct value of 29.3 a positive detection was reported for at least one tryptic peptide of VME1 (**Fig 5C**; sample #4, peptide EITVATSR). When the contaminating peak that was incorrectly assigned as the b3 fragment ion by Skyline is removed from the chromatograms, the pattern closely resembles that of its AQUA counterpart peptide.

Finally, we tested two different experimental procedures to obtain higher sensitivity and to decrease the overall LC-MS analysis time. To increase the measurement sensitivity, high pH reversed phase fractionation was applied to tryptic digests of clinical specimens. Fractionated peptides were collected in eight fractions, which were separately analyzed by PRM MS. This leaded in many cases to improved peptide detection and higher quantitation values, as exemplified in **Fig 6A** for several representative target peptides (left panels: unfractionated digests, right panels: fractionated digests). Peptide abundances were up to five times higher in the fractionated samples, while absolute quantitation based on comparison to estimated spiked-in amounts of AQUA counterpart peptides revealed that SARS-CoV-2 peptides could be detected in the low to mid-attomolar range. Shorter LC-MS gradient (20 min) resulted in overall slightly less identifications and quantitation results. Still, extremely low abundant target peptides could be reliably identified and quantified, despite the increased presence of contaminating peaks that are most likely the result of more crowded mass spectra (**Fig 6B**).

In conclusion, more sensitivity could be obtained by fractionation of tryptic digests prior to PRM analysis, although at the costs of longer analysis times. Shorter LC gradients were used to decrease the overall sample analysis time. While some peptides fell below the detection limit, the far majority of target peptides could still be reliably identified and quantified, also in samples of relatively high Ct values.

## Conclusions

We show that proteolytic peptides of SARS-CoV-2 proteins can be detected down to the mid-attomole range by targeted mass spectrometry. Our rough calculations indicate that the level of sensitivity should be sufficient to detect protein amounts corresponding to 1.2E7 copies. In addition, we have shown that the current sensitivity of PRM targeted mass spectrometry is

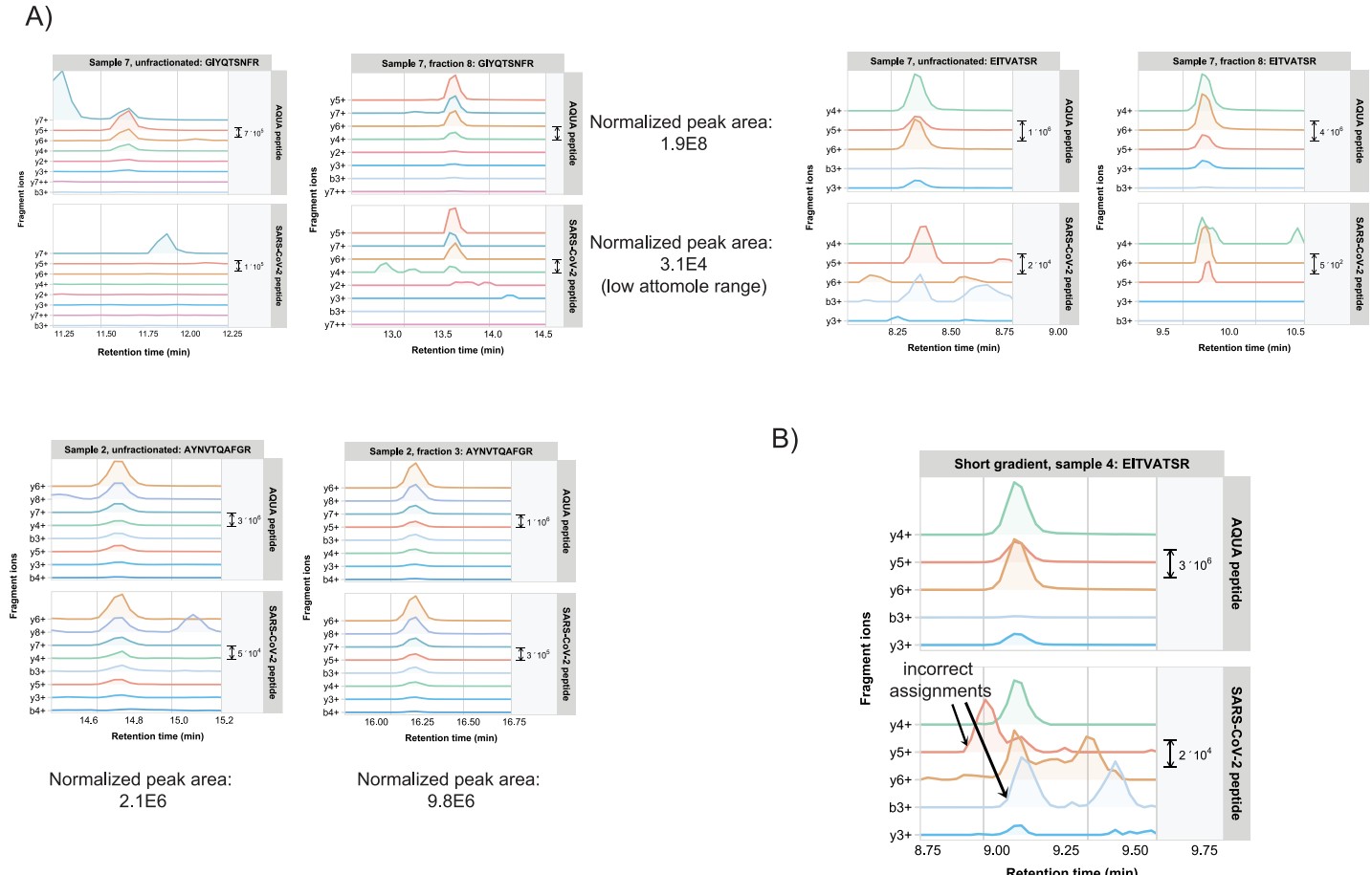

**Fig 6.** A) Comparison of one-shot versus high pH fractionation LC-MS PRM for several target peptides. For peptide GFYQTSNFR in Sample #7 the normalized peak area would correspond to the low attomolar range. B) Example of a positive target peptide identification in a 20 min gradient LC-MS run of a sample of high Ct value.

sufficiently high for the detection of virus proteins, in particular NP, present in patient material such as nasopharyngeal swabs and sputum. The identification of SARS-CoV-2 tryptic peptides was confirmed in an assay using AQUA synthesized heavy isotope labeled peptides spiked in as a positive control. Since we did not detect all SARS-CoV-2 tryptic peptides in every clinical sample that was positively tested for COVID-19 by PCR, the success of mass spectrometry based methods may depend on both the total absolute amount of viral proteins present in such samples as well as on the specific type of clinical specimens and the preparation thereof. Larger sample cohorts need to be included in future studies to further look into this.

PRM sensitivity in terms of numbers of detected virus particles is–as expected–not as high as that of RT-qPCR, which has been reported to be able to detect viral RNA in copy numbers as low as several 100s per reaction [17]. A major difference compared to conventional methods of viral diagnostics is that in this study proteins are analyzed as opposed to RNA in case of PCR based methods. This makes it an orthogonal detection method that could serve as a complementary tool for diagnosing SARS-CoV-2 infection.

The excellent label free quantitation capacity of targeted mass spectrometry over a wide concentration range makes this method particularly useful for *e.g.* the study of infection courses over time. By using spiked in AQUA peptides it should be possible to absolutely quantitate viral proteins, which would allow for the accurate monitoring of SARS-CoV-2 protein

abundances in *e.g.* time series. This could be useful in studies to the course of infection and for solving open questions on the importance of viral load in COVID-19 spreading.

Although several reports describing SARS-CoV-2 protein detection in clinical specimens have been published recently, we have investigated the relationship between PCR Ct values and mass spectral intensities in different independent patient cohorts without the need for immunopurification of SARS-CoV-2 proteins prior to mass spectrometry analysis. We observed an inverse linear relationship between the $^{10}$Log transformed summed AUCs of the fragment ion chromatograms and the PCR Ct value, which makes sense because of the logarithmic nature of the Ct value scale. Obviously, the number of data points is only limited in our case and the strength of this relationship is expected to become stronger with an increasing number of data points from larger cohorts. Factors that may contribute to the spread of the data points include the heterogeneity of the samples and differences in sample collection. Alternatively, the imperfect correlation may also reflect the nature of the samples. It is possible that both RNA and protein are present outside of infectious viral particles: RNA could be present without surrounding protein shell, while proteins or protein assemblies from disintegrated particles may still be floating around after infection. In such cases, a lower degree of correlation could be expected. Whether or not it is possible to use the developed methodology to differentiate between infectious virus particles and viral residue should be investigated, preferably in larger cohorts.

In conclusion, the current level of sensitivity of PRM proteomics methodology and the successful detection of SARS-CoV-2 proteins in patient material opens up ways to explore the use of mass spectrometry as a technology for clinical and diagnostics labs to detect viral infection in clinical specimens. Subsequent steps should now be focused on the optimization of fast sample preparation procedures and LC-MS throughput.

## Supporting information

**S1 Fig.** MS/MS spectra of tryptic peptides A) GFYAEGSR, B) ADETQALPQR and C) EITVATSR. Data visualization in PDV proteomics viewer (pdv.zhang-lab.org).
(PPTX)

**S2 Fig.** Tryptic peptide coverage in light green of A) Nucleocapsid (NCAP_SARS2) and B) Membrane protein (VME1_SARS2). Data visualization in PD2.3.
(PPTX)

**S3 Fig. PRM results from the SARS-CoV-2 infected Vero E6 cell lysates.** Fragment ion chromatograms for each of the Top5 or Top6 fragment ions are shown in different colors in a dilution series for tryptic peptides A) ADETQALPQR (NCAP_SARS2) and B) EITVATSR (VME1_SARS2). C) Library peptide fragmentation spectra for the indicated peptides. D) Calibration bar graphs for three target peptides.
(PPTX)

**S4 Fig.** A) PRM chromatograms of SARS-CoV-2 Nucleocapsid and VME1 tryptic peptides AYNVTQAFGR and VAGDSFAAYSR in four additional COVID-19 patient sputum specimens (#s 3–6) and one specimen from a patient infected with influenza B serving as a negative control (# 3). Chromatograms for each of the Top6 fragment ions are shown in different colors. The upper panels show the fragment ion chromatograms of the corresponding synthetic AQUA peptide AYNVTQAFG[R] (*m/z* 568.79) and VAGDSFAAYS[R] (*m/z* 605.79). S3 File contains the output in table format, including Skyline library dot product and total area fragment values. B) The corresponding Ct values for the sputum and throat swab samples from

PCR assays.
(PPTX)

**S5 Fig. PRM fragment ion chromatograms for various SARS-CoV-2 target peptides in one representative clinical specimen from patient cohort 1.**
(PPTX)

**S6 Fig. PRM fragment ion chromatograms for various SARS-CoV-2 target peptides in one representative clinical specimen from patient cohort 2.**
(PPTX)

**S1 File. Mascot/PD2.3 results of fractionated SARS-CoV-2 infected VeroE6 cell lysate.**
(XLSX)

**S2 File. PRM isolation *m/z* list.**
(XLSX)

**S3 File. Skyline export files for calibration curve.**
(XLSX)

**S4 File. Skyline export files for COVID-19 patient sputum specimens.**
(XLSX)

**S5 File. Skyline export files for COVID-19 patient cohort 1.**
(XLSX)

**S6 File. Skyline export files for COVID-19 patient cohort 2.**
(XLSX)

**S7 File. Complete set of fragment ion chromatograms for all cohort 1 specimens.**
(PDF)

**S8 File. Complete set of fragment ion chromatograms for all cohort 2 specimens.**
(PDF)

**S9 File. Detailed settings and parameters for the DDA and PRM MS analyses and skyline settings.**
(XLSX)

## Author Contributions

**Conceptualization:** Karel Bezstarosti, Mart M. Lamers, Jeroen A. A. Demmers.

**Data curation:** Mart M. Lamers, Jeroen A. A. Demmers.

**Formal analysis:** Jeroen A. A. Demmers.

**Funding acquisition:** Jeroen A. A. Demmers.

**Investigation:** Karel Bezstarosti, Mart M. Lamers, Bart L. Haagmans, Jeroen A. A. Demmers.

**Methodology:** Mart M. Lamers, Bart L. Haagmans, Jeroen A. A. Demmers.

**Project administration:** Jeroen A. A. Demmers.

**Resources:** Mart M. Lamers, Peter C. Wever, Khoa T. D. Thai, Jeroen J. A. van Kampen, Bart L. Haagmans, Jeroen A. A. Demmers.

**Software:** Wouter A. S. Doff, Jeroen A. A. Demmers.

**Supervision:** Jeroen A. A. Demmers.

**Validation:** Wouter A. S. Doff, Jeroen A. A. Demmers.

**Visualization:** Karel Bezstarosti, Wouter A. S. Doff, Jeroen A. A. Demmers.

**Writing – original draft:** Jeroen A. A. Demmers.

**Writing – review & editing:** Karel Bezstarosti, Jeroen A. A. Demmers.

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
