## [Decision Letter · Decision Letter 0]

23 Jun 2021

PONE-D-21-13444

Targeted proteomics as a tool to detect SARS-CoV-2 proteins in clinical specimens.

PLOS ONE

Dear Dr. Demmers,

Thank you for submitting your manuscript to PLOS ONE. After careful consideration, we feel that it has merit but does not fully meet PLOS ONE’s publication criteria as it currently stands. Therefore, we invite you to submit a revised version of the manuscript that addresses the points raised during the review process.

ACADEMIC EDITOR: Please address the comments raised by reviewers. 

We look forward to receiving your revised manuscript.

Kind regards,

Prasenjit Mitra, MD, MRSB, MIScT, FLS, FACSc, FAACC

Academic Editor

PLOS ONE

Journal Requirements:

2. Please provide additional details regarding participant consent. In the ethics statement in the Methods and online submission information, please ensure that you have specified (1) whether consent was informed and (2) what type you obtained (for instance, written or verbal, and if verbal, how it was documented and witnessed). If the need for consent was waived by the ethics committee, please include this information.

3. Please include your tables as part of your main manuscript and remove the individual files. Please note that supplementary tables (should remain/ be uploaded) as separate "supporting information" files

4.  Thank you for stating the following in the Financial Disclosure section:

We note that one or more of the authors are employed by a commercial company: "Star-shl Diagnostic Laboratories,"

Reviewers' comments:

Reviewer's Responses to Questions

**Comments to the Author**

1. Is the manuscript technically sound, and do the data support the conclusions?

Reviewer #1: Yes

2. Has the statistical analysis been performed appropriately and rigorously? 

Reviewer #1: Yes

3. Have the authors made all data underlying the findings in their manuscript fully available?

Reviewer #1: Yes

4. Is the manuscript presented in an intelligible fashion and written in standard English?

Reviewer #1: Yes

5. Review Comments to the Author

Reviewer #1: The manuscript, “Targeted proteomics as a tool to detect SARS-CoV-2 proteins in clinical specimens” by Bezstarosti et al, is an interesting and informative article. The paper addresses a timely question regarding the COVID-19 pandemic which has devastated the whole world. In this paper, the authors have done comprehensive research to establish a PRM MS-based method that can be utilized for COVID-19 diagnosis. Authors have cultured Vero E6 cells and infected them with SARs-CoV-2 to optimize the MS-based method which they have eventually extended to detect SARs-CoV-2 peptides in COVID-19 positive patient specimens. The authors were able to detect various SARS-CoV-2 tryptic peptides in nasopharyngeal swab samples of COVID-19 positive patients and have also investigated the relationship between amounts of viral RNA and protein abundances.

Authors are requested to address the following queries:

1) The study includes human sputum and swab samples. Did the authors get consent from the patients? Was this study been approved by any ethical committee? Please share the ethical committee documents and ICF used for the study. Please mention in the sample collection details of the methodology.

2) Authors have selected three peptides while working with Vero E6 cells infected with SARS-CoV-2 for PRM based targeting to diagnose COVID-19. Were there any specific criteria used to select these peptides? What other peptides were tested which have the potential to be used for diagnostic purposes.

3) In human sputum and swab samples, the authors performed PRM analysis on tryptic peptide AYNVTQAFGR and VAGDSGFAAYSR. Were these peptides analyzed in the first PRM experiment also where authors used three peptides from SARS-CoV-2 infected Vero E6 cells?

4) In the main text, while explaining figure 3 it is not clear what is panel C. There is an upper panel that shows AQUA peptide, the lower panel shows the corresponding endogenous peptide. Please clarify the text in the manuscript. The figure legend is explained correctly though.

5) While generating the calibration curve, peptide VAGDSGFAAYSR has been quantified in both sputum and swab samples. However, in nasopharyngeal swabs of cohort 1, the peptide has not been identified in any of the samples as depicted in supplementary information 1. Please explain.

6) The authors propose that higher sensitivity can be achieved by fractionating the samples. Since time is the limiting factor when one works with fractionated samples, what could be the least number of fractions that can be considered to improve the sensitivity of PRM based diagnosis.

7) Authors have found correlation between PCR Ct values and summed mass spectral intensities of peptides from PRM analysis. The current standard for Covid-19 positivity is Ct value of less than 35. Is there any absolute value of peptide quantification that can be used as a threshold for positivity of infection or is it just identification of the peptides which can confirm the presence of SARs-CoV-2 virus?

8) PRM analysis has been carried out using two cohorts of nasopharyngeal swab samples from Covid-19 positive patients in the current study. To make the PRM based diagnostic method feasible both in terms of analysis and time, authors should summarize the list of peptides that can serve as potential candidates to detect the presence of SARS-CoV-2 virus in nasopharyngeal swabs.

6. PLOS authors have the option to publish the peer review history of their article (what does this mean?). If published, this will include your full peer review and any attached files.

Reviewer #1: **Yes: **Aditi Chatterjee

---

## [Author Response · Author response to Decision Letter 0]

15 Jul 2021

For detailed replies to all questions, suggestions and comments raised by the editor and/or the reviewers, please see the file 'Response_to_Reviewers'.

---

## [Editor Report · Decision Letter 1]

14 Oct 2021

Targeted proteomics as a tool to detect SARS-CoV-2 proteins in clinical specimens.

PONE-D-21-13444R1

Dear Dr. Demmers,

We’re pleased to inform you that your manuscript has been judged scientifically suitable for publication and will be formally accepted for publication once it meets all outstanding technical requirements.

Kind regards,

Prasenjit Mitra, MD, CBiol, MRSB, MIScT, FLS, FACSc, FAACC

Academic Editor

PLOS ONE
---

## [Editor Report · Acceptance letter]

28 Oct 2021

PONE-D-21-13444R1 

Targeted proteomics as a tool to detect SARS-CoV-2 proteins in clinical specimens. 

Dear Dr. Demmers:

I'm pleased to inform you that your manuscript has been deemed suitable for publication in PLOS ONE. Congratulations! Your manuscript is now with our production department. 

Kind regards, 

on behalf of

Dr. Prasenjit Mitra 

Academic Editor

PLOS ONE